# Fallow Deer (*Dama dama*) as a Reservoir of Shiga Toxin-Producing *Escherichia coli* (STEC)

**DOI:** 10.3390/ani10050881

**Published:** 2020-05-19

**Authors:** Anna Szczerba-Turek, Bernard Kordas

**Affiliations:** Department of Epizootiology, Faculty of Veterinary Medicine, University of Warmia and Mazury in Olsztyn, Oczapowskiego 13, 10-718 Olsztyn, Poland; bernard.kordas@student.uwm.edu.pl

**Keywords:** fallow deer (Dama dama), Shiga-like toxin-producing *E. coli* (STEC), food-borne zoonoses, Poland

## Abstract

**Simple Summary:**

Infections caused by Shiga toxin-producing *Escherichia* (*E.*) *coli* (STEC) are the third most typically reported zoonosis within the European Union after campylobacteriosis and salmonellosis. STEC pathogens are responsible for the outbreaks of serious diseases in humans, including haemolytic uraemic syndrome (HUS), haemorrhagic colitis (HC) and diarrhoea (D). Red deer, roe deer and wild boars are important environmental reservoirs of foodborne pathogens that may cause serious diseases in humans and contaminate fresh food products. The occurrence of STEC and attaching and effacing (AE)-STEC in the Polish population of wild fallow deer was analysed in this study. The presence of potentially pathogenic STEC/AE-STEC in fallow deer poses a risk of carcass contamination, which could have serious consequences because venison can also be consumed raw as carpaccio or steak tartare. Only a few reports on wild animals as a reservoir of foodborne pathogens in European countries have been published to date, and the present study attempts to fill in this knowledge gap by assessing the possible epidemiological risk related to STEC/AE-STEC in fallow deer. Three isolates had the virulence profile that is associated with HUS/D/HC according to the FAO/WHO report. The results of this study suggest that fallow deer are carriers of STEC/AE-STEC that are potentially pathogenic to humans.

**Abstract:**

Shiga toxin-producing *Escherichia* (*E.*) *coli* (STEC) are responsible for the outbreaks of serious diseases in humans. Only a few reports on fallow deer as a reservoir of foodborne pathogens have been published to date. The purpose of this study was to determine the occurrence of STEC strains in the fallow deer population in Poland. In all, 94 fallow deer swabs were tested. Polymerase chain reaction (PCR) was performed to detect the virulence profile of *stx1*, *stx2* and *eae* or *aggR* genes, to identify the subtypes of *stx1* and *stx2* genes and to perform O and H serotyping. STEC and attaching and effacing (AE)-STEC were identified in 13 isolates (13.83%). The most hazardous virulence profile was detected in three strains, namely *stx2d* serotype O103:HNM, *eae/stx1a* serotype O26:HNM and *eae/stx1a* serotype O157:H7. The predominant *stx* gene was *stx2*, which was identified in 76.92% of isolates. *E. coli* O157 was detected in 4/94 (4.26%). Other *E. coli* serogroups, O26, O103, O111 and O145, were identified in 14/94 fallow deer (14.89%). The present findings suggest that fallow deer are carriers of STEC/AE-STEC that are potentially pathogenic to humans.

## 1. Introduction

Shiga toxin-producing *Escherichia* (*E.*) *coli* (STEC) are generally recognised as a significant cause of foodborne diseases, such as haemolytic uraemic syndrome (HUS), haemorrhagic colitis (HC) and diarrhoea (D) [1]. In 2018, 8161 confirmed cases of STEC infections were registered in the European Union (EU) [2], and the number of cases and the notification rate of STEC increased significantly from 2017 (n = 6073), which made STEC the third most frequently reported zoonosis in the EU after campylobacteriosis (n = 246,571) and salmonellosis (n = 91,857) [2,3]. Between 2009 and 2018, the prevalence of human STEC infections increased steadily, principally due to a large STEC outbreak in 2011 [2,4]. The observed increase can be partially attributed to higher detection rates following the STEC outbreak, as well as advanced laboratory methods, including polymerase chain reaction (PCR), for direct extraction of bacterial DNA from specimens and strain characterisation [2]. According to the Food and Agriculture Organization (FAO) of the United Nations and the World Health Organization (WHO), the pathogenicity of STEC to humans depends on the occurrence of selected virulence factors, in particular Shiga toxin 1 (*stx1*)*,* Shiga toxin 2 (*stx2*), attaching and effacing *E. coli* (*eae*), or a transcriptional activator of aggregative adherence fimbria I (*aggR*) [1]. Shiga toxin genes are subdivided into three *stx1* subtypes, namely *a, c* and *d*, and seven *stx2* subtypes, namely *a*, *b*, *c*, *d*, *e*, *f* and *g* [5]. The latest studies have demonstrated that STEC strains classified as subtype *stx2a* and possessing adherence genes *eae* or *aggR* pose the most serious health risk to humans and have the highest potential to induce HUS [1,6,7]. According to the strategy for assessing health risks based on an analysis of STEC virulence genes, strains containing *stx2a* or *stx2d* and *eae* or *aggR* genes are most harmful to humans. Such strains have the highest capacity to cause D, HC and HUS. In D and HC patients, strains with the *stx2c/eae* or *stx1a/eae* virulence profiles have been identified [1]. Most STEC infections are caused by the ingestion of faecal-contaminated food or water, indirect or direct exposure to animal vectors, or secondary transmission from humans to humans [8]. Household ruminants, especially cattle, are recognised as the major natural STEC reservoir [8]. Large wildlife, such as deer (*Cervus elaphus*), roe deer (*Capreolus capreolus*) and wild boars (*Sus scrofa*), can also be healthy hosts of O157:H7 and non-O157 STEC, but the number of publications on the virulence profiles of strains isolated from these animals is not significant [9,10,11,12,13,14,15,16]. According to the European Food Safety Authority (EFSA) and the European Centre for Disease Prevention and Control (ECDC) [2], both the number of reports and the number of tested animal samples have decreased steadily in recent years. Animal samples are still frequently assayed with the use of techniques that reveal only *E. coli* O157, which puts into question the reliability of the reports on STEC detection in animals in the EU. In our previous study, 21 rectal swabs from fallow deer were analysed, and the occurrence rate of STEC/AE-STEC strains in the fallow deer population in Poland was determined at 9.52%. The prevalence of enteropathogenic *E. coli* (EPEC) reached 33.33%, and the prevalence of the “top five” serogroups was determined at 9.52%. One strain with the *stx1a/stx2g* virulence profile and one with *stx2b* were identified. However, the results of that study could not be used to formulate reliable conclusions due to the small number of rectal swabs (21 samples) [15]. In the present study, additional swabs collected during the 2017–2018 and 2018–2019 hunting seasons were analysed to identify virulence profile, *stx1* and *stx2* subtypes and the prevalence of O-serogroups in STEC obtained from the Polish population of fallow deer (*Dama dama*).

## 2. Materials and Methods

### 2.1. Sampling

During two autumn–winter hunting seasons of 2017–2018 and 2018–2019 (from 1 September to the end of February for males and fawns, and from 1 September to 15 January for females) [17,18] in Poland, a total of 94 rectal swabs were obtained from fallow deer (*Dama dama*). The swabs were collected in collaboration with hunters. Unfortunately, nothing is known about the health status of the animals from which the swabs were taken. The swabs were collected from each animal before evisceration, placed in tubes and transported refrigerated to the laboratory within 48 h. All swabs were collected as part of a standard procedure; therefore, ethics approval was not required.

### 2.2. Detection of STEC Strains and stx1 and stx2 Subtypes

Swabs were mixed in 5 mL of buffered peptone water (BPW; BTL) in aseptic conditions and incubated for 20–24 h at 37 °C. DNA was extracted from 1 mL of culture using the Genomic Mini kit (A&A Biotechnology, Gdynia, Poland) following the manufacturer’s recommendations. The samples were screened for the presence of *stx1*, *stx2*, *eae* and *aggR* genes following the protocol proposed by the European Union Reference Laboratory for *E. coli* (EU-RL VTEC_Method 01 for *E. coli*) and the method described by Schmidt et al. [19,20,21,22]. In order to isolate a single STEC/AE-STEC strain, all samples that tested positive for the analysed genes in PCR were treated as previously described by Szczerba-Turek et al. [15]. The *stx1* and *stx2* subtypes were determined following the protocol developed by Scheutz et al. [5] and the protocol proposed by the European Union Reference Laboratory for *E. coli* (EU-RL VTEC_Method 006) [23].

### 2.3. Serotyping by Polymerase Chain Reaction (PCR)

The STEC/AE-STEC/EPEC serotypes were determined by the PCR assay based on the EU-RL protocol for *E. coli* (EU-RL VTEC_Method 003) [24,25], for O-antigen-encoding genes (wzx)—O26, O103, O111, O145 and O157—and the protocols proposed by Durso et al., Gannon et al. and Mora et al. [26,27,28] for H antigens encoding the *fliC* gene (specific to flagellar genes)—H7, H8, H11, H21 and H28. All PCR amplifications were performed with the HotStartTaq Plus DNA Polymerase Kit (Qiagen, Venlo, Netherlands) and the HotStartTaq Plus Master Mix Kit (Qiagen), according to the manufacturer’s recommendations. The annealing temperature for every primer was described previously [15]. The PCR products were separated by electrophoresis in 2% agarose gel with the Midori Green Advanced DNA Stain (Nippon Genetics Europe GmbH, Düren, Germany).

### 2.4. Statistics

For basic statistical analysis, the binomial (Clopper–Pearson) “exact” method based on beta distribution at a significance level of α = 0.05 and 95% confidence interval was applied. All statistical analyses were performed with free EpiTools epidemiological calculators (http://epitools.ausvet.com.au) [29].

## 3. Results

In the studied population of 94 fallow deer (*Dama dama*), STEC were identified in four isolates (4.26%, 95% CI = 1.17–10.54), AE-STEC were detected in nine isolates (9.57%, 95% CI = 5.99–19.97), and EPEC were identified in eight isolates (8.51%, 95% CI = 3.75–16.08). STEC/AE-STEC were identified in 13 isolates (13.83%, 95% CI = 7.57–22.49). In the examined group of STEC/AE-STEC, *stx2* was the predominant gene and was identified in 10 isolates (76.92%, 95% CI = 46.19–94.96), whereas the *stx1* gene was detected in seven isolates (53.85%, 95% CI = 25.13–80.78). Three strains harboured the *stx1a* gene, one strain harboured the *stx2a* gene, one strain harboured the *stx2d* gene and four strains harboured the *stx2b* gene. The occurrence of *stx1* and *stx2* subtypes is shown in Table 1. *E. coli* O157 strains were detected in 4/94 rectal swabs from fallow deer (*Dama dama*) (4.26%, 95% CI = 1.17–10.54) and three of them were identified as O157:H7 (3.19%, 95% CI = 0.66–9.04). *E. coli* O26, O103, O111 and O145 were identified in 14/94 fallow deer (14.89%, 95% CI = 8.391–23.72). The “top five” were identified in 18/94 samples (19.15%, 95% CI = 11.76–28.56). The results of serotyping are presented in Table 1.

## 4. Discussion

According to the concept of “One Health” developed in the early 2000s, human health and animal health (livestock, pets and wildlife) are interdependent and bound to the health of the ecosystems in which they exist, which is why their overall health status should be closely monitored [30,31,32]. Human infections with STEC have been reported after the consumption of deer meat or the meat of other game animals [33,34,35,36,37,38,39]. This study analysed the occurrence of STEC/AE-STEC in fallow deer that can be an important natural reservoir of STEC/AE-STEC strains. The above has significant implications for public health because the number of fallow deer farms has increased due to a decline in the profitability of livestock farming. The global population of farmed deer is estimated at nearly 5 million [40]. In Europe, the highest number of fallow deer breeding farms are located in Germany and the United Kingdom [41], but other European countries such as Sweden, Czechia, Austria, Spain, Italy, Switzerland, Norway, Slovakia, Hungary and Poland have well-established traditions relating to deer breeding, venison production and consumption as well as game hunting [42]. Fallow deer are reared in Europe on account of their long lifespan and their ability to withstand disease and low temperatures in winter and produce high-quality meat that is valued for its unique taste and high nutritional value [42,43,44,45]. Fallow deer are hunter-harvested for meat, skin and antlers. The fact that STEC/AE-STEC strains that are potentially pathogenic to humans were present in swabs collected from free-living fallow deer points to the possibility of environmental contamination, particularly in grasslands grazed by cattle, sheep and other farm animals, which may result in the cross-transmission of STEC/AE-STEC between different animal species. Such a situation was observed in the USA where potentially pathogenic STEC/AE-STEC strains derived from deer were found in cows and their cross-transmission was reported [46]. In addition, STEC/AE-STEC have also been isolated from water bodies and vegetable crops, posing a real risk to human health. Therefore, it is important to monitor free-living animals for the occurrence of zoonotic strains that are potentially pathogenic to humans. STEC should also be classified based on type (*stx1* or *stx2*). The prevalence of STEC/AE-STEC that are potentially pathogenic to humans in the fallow deer population is an important consideration. Hunters have to comply with the regulations laying down specific hygiene rules for food of animal origin and specific rules for the organisation of official controls on products of animal origin intended for human consumption [47,48].

There are no previously published reports of the occurrence of STEC/AE-STEC, containing *stx1* and *stx2* subtypes, and the virulence genes of strains isolated from fallow deer. In our previous study, which analysed only 21 faecal swabs from the Polish population of fallow deer, the occurrence of STEC/AE-STEC was estimated at 9.52%, whereas *E. coli* O157 was not detected [15]. In the present investigation, the occurrence of STEC/AE-STEC was estimated at 13.8% (13/94), the occurrence of *E. coli* O157 strains was estimated at 4.2% (4/94) and three strains were identified as O157:H7 (3.2%) (3/94). One EPEC strain had the *eae/stx1a* virulence profile, whereas the *eae/stx1NS* profile was determined in two AE-STEC strains. Three isolates had the virulence profile that is associated with HUS/D/HC according to the FAO/WHO report [1], namely *stx2d* serotype O103:HNM, *eae/stx1a* serotype O26:HNM and *eae/stx1a* serotype O157:H7.

A total of 115 samples were analysed in the current study and our previous research [15]. The occurrence of STEC/AE-STEC was estimated at 13% (15/115), the occurrence of *E. coli* O157 strains was estimated at 3.48% (4/115) and three strains were identified as O157:H7 (2.6%) (3/115). The prevalence of EPEC strains was determined at 6.1% (7/115). The predominant *stx* subtype was *stx2* that was identified in 10.43% of the samples (12/115), whereas subtype *stx1* was found in 6.95% of the analysed swabs (8/115). The presence of potentially pathogenic STEC/AE-STEC in fallow deer poses a risk of carcass contamination, which could have serious consequences because venison can also be consumed raw as carpaccio or steak tartare. For this reason, the microbiological safety of meat from fallow deer has to be ensured. However, STEC that are potentially pathogenic to humans were found in the examined samples, which points to the risk of carcass and environmental contamination.

## 5. Conclusions

Fallow deer can carry STEC/AE-STEC that are potentially pathogenic to humans. Three strains isolated in this study had the virulence profile associated with human infections, namely *stx2d* serotype O103:HNM, *eae/stx1a* serotype O26:HNM and *eae/stx1a* serotype O157:H7. This is an important finding that indicates that high hygiene standards have to be observed in the process of dressing fallow deer carcasses in the field and preparing fallow deer meat. The occurrence of STEC/AE-STEC in the rectal swabs of wild fallow deer should also be taken into account when developing strategies aiming to limit and/or control this pathogen in water, livestock, pets and wildlife. This is the first complete report describing the virulence of STEC/AE-STEC isolated from fallow deer.

## Figures and Tables

**Table 1 animals-10-00881-t001:** Pathotypes and O serogroups of Shiga toxin-producing *Escherichia coli* / attaching and effacing (STEC/AE)-STEC and enteropathogenic *Escherichia*
*coli* (EPEC) isolated from fallow deer (*Dama dama*).

Virulence Genes	Number of Samples	Number of Strains	*stx1* Subtype (Number)	*stx2* Subtype (Number)	Serogroup (Number)	Time of Collection Year/Month
STEC *stx1*	0	--	--	--	--	
STEC *stx2*	4	2		*stx2a* (1) *stx2d* (1)	ONT:HNM (1) O103:HNM (1)	2019/Jan. 2018/Nov
STEC *stx1 stx2*	2	2	*stx1a* (1) *stx1NS* (1)	*stx2NS* (1) *stx2b* (1)	ONT:H7 (1) ONT:HNM (1)	2019/Jan. 2018/Dec.
AE-STEC *stx1 eae*	8	3	*stx1a* (2) *stx1NS* (1)		O26:HNM (1), O157:H7 (1) O157:H7 (1)	2018/Oct., 2018/Nov. 2018/Nov.
AE-STEC *stx2 eae*	11	4		*stx2b* (2) *stxNS* (2)	O103:HNM (2) O103:HNM (1), O26:HNM (1)	2017/Nov., 2018/Oct. 2018/Feb., 2018/Dec.
AE-STEC *stx1 stx2 eae*	4	2	*stx1NS* (1) *stx1NS* (1)	*stx2b* (1) *stxNS* (1)	O103:HNM (1) O26:HNM (1)	2019/Jan. 2018/Oct.
EPEC *eae*	34	8			O26:HNM (1), O103:HNM (2), O145:H7 (3), O157:H7 (1), O157:HNM (1)	2018/Dec., 2018/Oct., 2018/Oct. 2018/Jan. (2), 2018/Feb., 2019/Jan. 2018/Nov.
Total	63	21	7	10	21	

NS, non-subtype; HNM, non-motile H antigen; ONT, untypable O antigen.

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
