# Peer review of "Fallow Deer (Dama dama) as a Reservoir of Shiga Toxin-Producing Escherichia coli (STEC)"

_animals, 2020, doi:10.3390/ani10050881_

Round 1
Reviewer 1 Report
The authors of “Fallow deer (Dama dama) as a reservoir of Shiga toxin-producing Escherichia coli (STEC)” have aimed to describe the occurrence of STEC in large wildlife that is fallow deer in relation to public health risks. A topic which of interest to the reader in the current One Health era.
The followed procedure regarding detection and isolation of the STEC/AE-STEC is suitable. But the authors do not address the fact that they only succeed to isolate STECs in 44.8% (13/29) of the samples PCR positive for stx1 or stx2. Please elaborate and discuss to improve the procedure.
Samples were collected in two winter seasons, however the authors do not describe from which year/period the STEC isolates originate. Please adapt.
Please do not start the results section with the description of the serotypes found, which include STECs/AE-STECs as well as EPECs, while the title of the manuscripts states STEC. This is confusion to a reader. Please give the EPEC results a less prominent role in the whole manuscript.
Please check the English language by a native speaker.
Minor comments
Line 54; Shiga toxin 1 and Shiga toxin 2 should not be written in italics only the gene names stx1 and stx2
Line 71; E. coli in italics
Line 83; Please replace 2 by two regarding the number of seasons
Line 84; because instead of becouse
Line 85; Please replace understood by known
Line 94; E. coli in italics
Line 96; strain
Line 101; E. coli in italics
Line 106; either The amplicons or The PCR products
Line 116-117; E. coli
Line 166; In addition instead of However
Line 170; please remove double space after AE-STEC
Author Response
13.052020
The Authors would like to thank the Editor and Reviewers for a thorough perusal of our manuscript, and for the valuable comments and suggestions that helped use improve the quality of the paper. Below are point-by-point responses to the comments made by Reviewer 1. All content-related changes introduced to the text are highlighted in yellow, and linguistic corrections are marked in red.
Reviewer #1
Comment 1:
The followed procedure regarding detection and isolation of the STEC/AE-STEC is suitable. But the authors do not address the fact that they only succeed to isolate STECs in 44.8% (13/29) of the samples PCR positive for stx1 or stx2. Please elaborate and discuss to improve the procedure.
Author’s response:
We are grateful for this suggestion, below is the step by step procedure for isolating a single colony:
- Rectal swabs were placed in 5 ml of BPW at 37oC for 24 h,
- DNA was isolated from 1 ml of cultured bacteria using the Genomic Mini kit (A&A Biotechnology, Poland), according to the manufacturer’s recommendations
- 3. Genes were detected using Paton/Paton with QiagenHotStart Taq Plus polymerase with 4 pairs of primers according to the procedure http://old.iss.it/binary/vtec/cont/EU_RL_VTEC_Method_01_Rev_0.pdf
- If any of the products eae -384 bp, vtx1-180bp, vtx2-255bp or vtx2f – 428bp were obtained, the procedure of isolating a single colony was carried out
- 100 µl of each enrichment culture was streaked on MacConkey agar (MC) plates and incubated at 41oC for 24 h
- 1µl loop from the confluent growth of a single colony (from MC) was plated on Tryptic Soy Agar (TSA) and incubated at 37oC for 24h,
- A bacterial layer of TSA was diluted with water and treated at 100oC for 10 min., and used as a template for Paton/Paton PCR
- If any of the 4 genes was obtained after PCR, bacterial culture was taken from the same line (a bacterial layer of TSA), transferred to Tryptone Soya Broth (TSB) and left overnight
- DNA was isolated using the Genomic Mini kit (A&A Biotechnology) and put Paton/Paton PCR was carried out
- A minimum of 50 single colonies from MC were checked, and the procedure was discontinued if the target colony was not found
- In some cases, STEC were found immediately, whereas in other cases it seemed to be impossible
We would be grateful for any suggestions on how to improve the above procedure. This is a very difficult problem because the results may vary widely. Only a few authors have shown their results after the first screening. In our previous study with wild boars, we obtained 78 potential STEC/AE-STEC isolates after the first screening by PCR, but ultimately we identified only 43 strains, i.e. 55%. We are afraid that this is the actual efficiency of this methodology the Authors describe this part in next article about foxes because this may be a good part of discussion
Szczerba-Turek, A.; Socha, P.; Bancerz-Kisiel, A.; Platt-Samoraj, A.; Lipczynska-Ilczuk, K.; Siemionek, J.; Konczyk, K.; Terech-Majewska, E.; Szweda, W. Pathogenic potential to humans of Shiga toxin-producing Escherichia coli isolated from wild boars in Poland. Int. J. Food Microbiol. 2019, 300, 8-13.
Comment 2:
Samples were collected in two winter seasons, however the authors do not describe from which year/period the STEC isolates originate. Please adapt.
Author’s response:
The year and month have been included in the Table.
Comment 3:
Please do not start the results section with the description of the serotypes found, which include STECs/AE-STECs as well as EPECs, while the title of the manuscript’s states STEC. This is confusion to a reader. Please give the EPEC results a less prominent role in the whole manuscript.
Author’s response:
As suggested by the Reviewer, this passage has been rewritten, and the description of EPEC results has been modified
Comment 4:
Line 54; Shiga toxin 1 and Shiga toxin 2 should not be written in italics only the gene names stx1 and stx2.
Author’s response:
The relevant corrections have been made - “Shiga toxin 1 (stx1), Shiga toxin 2 (stx2)”
Comment 5:
Line 71; E. coli in italics
Author’s response:
“E. coli” has been italicized - “E. coli”
Comment 6:
Line 83; Please replace 2 by two regarding the number of seasons
Author’s response:
“2” has been replaced with “two”
Comment 7:
Line 84; because instead of because
Author’s response:
“becouse” has been replaced with the correct word “because”
Comment 8:
Line 85; Please replace understood by known
Author’s response:
“understood” has been replaced with “known”
Comment 9:
Line 94; E. coli in italics
Author’s response:
“E. coli” has been italicized - “E. coli”
Comment 10:
Line 96; strain
Author’s response:
“strains” has been replaced with “strain”
Comment 11-15:
Line 101; E. coli in italics
Line 106; either The amplicons or The PCR products
Line 116-117; E. coli
Line 166; In addition instead of However
Line 170; please remove double space after AE-STEC
Author’s response:
The relevant corrections have been made.

Reviewer 2 Report
This is an interesting artcile that provides new and important insights into the carrier state of reservoirs in wildife used for human consumption which can transmit proven zoonotic E. coli strain of high significance.
I recommend minor English corrections in the text as well as further explanation on a few topics particularly regarding the legality of hunting and the fact that no ethical committee was needed, which may sound backwards in a time where at least the University commitee agrees with the project and it does not cause unecessary suffering to the animals. Another question is about the definition of EPEC, whereas the ones found where considered typical or atypical or if this hasn't been tested. Plus defining the overlapping betweeen livestock and wild deer as carriers/reservoirs for Stx/EPEEC foodborne infections to humans.
Lines:
12: English review of chargeable/great.
13: Sicentific names when first introduced and throughout the paper.
15: Definition of AE.
29: English review - work out. better to use determine or detect for instance.
32: definition of AE.
69-70: rephrasing: the number of reports decreased steadily in recent years but so did the number of tested animal samples.
84: Number of malesxfemales. All were adults?
84: because.
85: rephrase: thanks to the cooperation with hunters throughout this study.
85: no information about the health status of these carriers was previously known, but they appeared to be clinically healthy, showing no signs of external or internal pathological signs.
88: Provide additional information on the legality of hunting in Poland during this season and how this is regulated.
90: vortexed/mixed.
94: E. coli in italics.
126: were these thested also for distinguising between typical and atypical EPEC? if not describe in the methods section or if they are actually all atypical isolates/or not tested to distinguish.
140: However is there a possible common route of cross-transmission between reared livestock grazing in the same grounds as wild fallow deer? Such studies have been previously undertaken in areas such as the USA and Canada where wildlife carrying Stx/EPEC strains was possibly involved in the maintenance/cycle of colonisation of livestock and back to wildlife. It is important to further explain if the wild population of given regions in Europe is also a potential reservoir for free-ranging livestock.
148: There are no previously published...
161: define again if typical or atypical.
170: Summarise the initial conclusion phrase by removing: The conclusions of this examination showed that.
Author Response
13.052020
The Authors would like to thank the Editor and Reviewers for a thorough perusal of our manuscript, and for the valuable comments and suggestions that helped use improve the quality of the paper. Below are point-by-point responses to the comments made by Reviewer 1. All content-related changes introduced to the text are highlighted in yellow, and linguistic corrections are marked in red.
Reviewer #1
Comment 1:
I recommend minor English corrections in the text as well as further explanation on a few topics particularly regarding the legality of hunting and the fact that no ethical committee was needed, which may sound backwards in a time where at least the University commitee agrees with the project and it does not cause unecessary suffering to the animals
Author’s response:
Samples were taken during scheduled fallow deer hunts in Poland. The information on the fallow deer hunting season has been added to subsection 2.1 Sampling ref. 17 and 18, the samples were collected by courtesy of the hunters involved in the sampling procedure, which was not easy. The project did not include sampling because it had been conducted earlier by the hunters, the analyzed fallow deer would be hunter-harvested anyway, and the purchase of reagents for molecular and microbiological tests, and the publication of results were financed.
Comment 2:
Another question is about the definition of EPEC, whereas the ones found where considered typical or atypical or if this hasn't been tested.
Author’s response:
This has not been tested, Authors confirmed that they tested only aggR gene without bfpB gene, but this valuable suggestion will be taken into account in our future research
Comment 3:
Plus defining the overlapping betweeen livestock and wild deer as carriers/reservoirs for Stx/EPEEC foodborne infections to humans.
Author’s response:
A brief discussion regarding livestock and wild deer has been included in the revised manuscript.
Comment 4:
Line 12: English review of chargeable/great.
Author’s response:
The relevant corrections have been made
Comment 5:
Line 13; Sicentific names when first introduced and throughout the paper.
Author’s response:
Authors put “hemorrhagic colitis (HC) instead “bloody diarrhoea - (BD)” in the revised manuscript
Comment 6:
Line 15: Definition of AE.
Author’s response:
“AE-STEC“ has been replaced with “attaching and effacing (AE)-STEC ”
Comment 7:
Line 29: English review - work out. better to use determine or detect for instance.
Author’s response:
“work out” has been replaced with “detect”
Comment 8:
Line 32: definition of AE
Author’s response:
AE has been replaced with “attaching and effacing”
Comment 9;
Line 69-70: rephrasing: the number of reports decreased steadily in recent years but so did the number of tested animal samples.
Author’s response:
The phrase “the number of tested animal samples and the number of reports decreased steadily in recent years” has been removed and replaced with “both the number of reports and the number of tested animal samples have decreased steadily in recent years”.
Comment 10;
Line 84: Number of malesxfemales. All were adults?
Author’s response:
Unfortunately, the Authors do not have any information on this subject
Comment 11;
Line 84: because.
Author’s response:
“becouse” has been replaced with “because”
Comment 12;
Line 85: rephrase: thanks to the cooperation with hunters throughout this study.
Author’s response:
The above phrase has been modified
Comment 13;
Line 85: no information about the health status of these carriers was previously known, but they appeared to be clinically healthy, showing no signs of external or internal pathological signs.
Author’s response:
The animals were harvested during scheduled hunts, and they appeared to be clinically healthy without any pathological sings
Comment 14;
Line 88: Provide additional information on the legality of hunting in Poland during this season and how this is regulated.
Author’s response:
The following sentence has been added: “During two autumn-winter hunting seasons of 2017/2018 and 2018/2019 (from 1 September to the end of February for males and fawns, and from 1 September to 15 January for females)[17-18]
- Regulation of the Minister of the Environment of 16 March 2005 on the definition of hunting periods for wildlife, Polish Law Gazette no. 48. 459, 2005. Internet System of Legal Documents. Available online. http://prawo.sejm.gov.pl/isap.nsf/download.xsp/WDU20050480459/O/D20050459.pdf (in Polish). accessed April 30.2020]
- Regulation of the Minister of the Environment of 1 August 2017 amending the Regulation of the Minister of the Environment of 16 March 2005 on the definition of hunting periods for wildlife. Polish Law Gazette 1487, 2017. Internet System of Legal Documents. Available online.http://isap.sejm.gov.pl/isap.nsf/download.xsp/WDU20170001487/O/D20171487.pdf (in Polish) accessed April 30, 2020.
Comment 15;
Line 90: vortexed/mixed.
Author’s response:
“smashed” has been replaced with “mixed”
Comment 16;
Line 94: E. coli in italics.
Author’s response:
“E. coli” has been italicized
Comment 17;
Line 126: were these tested also for distinguising between typical and atypical EPEC? if not describe in the methods section or if they are actually all atypical isolates/or not tested to distinguish.
Author’s response:
We are aware that a distinction should be made between atypical EPEC (a-EPEC) and typical EPEC (the latter possessing the bfp-plasmid), bfp-plasmid, we tested only eggR gene without bfpB gene.
Comment 18;
Line 140: However is there a possible common route of cross-transmission between reared livestock grazing in the same grounds as wild fallow deer? Such studies have been previously undertaken in areas such as the USA and Canada where wildlife carrying Stx/EPEC strains was possibly involved in the maintenance/cycle of colonisation of livestock and back to wildlife. It is important to further explain if the wild population of given regions in Europe is also a potential reservoir for free-ranging livestock.
Author’s response:
Thank you for this valuable suggestion, the relevant passage has been added to the Discussion section in the revised manuscript.
Comment 19;
Line 148: There are no previously published...
Author’s response:
The relevant correction has been made
Comment 20;
Line 161: define again if typical or atypical.
This has not been tested
Comment 21
Line 170: Summarise the initial conclusion phrase by removing: The conclusions of this examination showed that.
Authors response
The above phrase has been removed

Round 2
Reviewer 1 Report
The authors of “Fallow deer (Dama dama) as a reservoir of Shiga toxin-producing Escherichia coli (STEC)” have improved their paper according to my review suggestions. Although I still miss the elaboration and discussion regarding the improvement of procedure. Due to providing a more detailed description of the method followed my suggestions would be;
- The EU_RL_VTEC_Method_02
- In instead of plating 100 µl of each enrichment culture first make a 10-fold dilution range from the enrichment and then plate 100 µl of the adequate dilutions.
- Use additional chromogenic agars to recognize E. coli more easily from other flora
Author Response
18.052020
The Authors would like to thank the Editor and Reviewer 1 for a thorough perusal of our manuscript, and for the valuable comments and suggestions that helped use improve the quality of the paper. Below are responses to second the comments made by Reviewer 1.
Reviewer #1
Comment 1:
The authors of “Fallow deer (Dama dama) as a reservoir of Shiga toxin-producing Escherichia coli (STEC)” have improved their paper according to my review suggestions. Although I still miss the elaboration and discussion regarding the improvement of procedure. Due to providing a more detailed description of the method followed my suggestions would be;
- The EU_RL_VTEC_Method_02
- In instead of plating 100 µl of each enrichment culture first make a 10-fold dilution range from the enrichment and then plate 100 µl of the adequate dilutions.
- Use additional chromogenic agars to recognize E. coli more easily from other flora
Author’s response:
The Author’s confirm that the article still lacks elaboration and discussion on improving the procedure of obtaining single STEC colonies, but according to the Author’s this is a serious problem and requires more time and, above all, checking the methods proposed, among others, by the reviewer. The Author’s are grateful for the suggested changes in the methodology and promise to apply them soon – especially Real-Time PCR according to The EU_RL_VTEC_Method_02, to improve the methodology and in the next article, there will be a part concerning this problem in a discussion. Once again, the Author’s are grateful for the review, which they hope will help them to improve their work with STEC.

This manuscript is a resubmission of an earlier submission. The following is a list of the peer review reports and author responses from that submission.